# Defects in Mitochondrial Functions Affect the Survival of Yeast Cells Treated with Non-Thermal Plasma

**DOI:** 10.3390/ijms24119391

**Published:** 2023-05-28

**Authors:** Anna Strížová, Paulína Šmátralová, Petra Chovančíková, Zdenko Machala, Peter Polčic

**Affiliations:** 1Department of Biochemistry, Faculty of Natural Sciences, Comenius University in Bratislava, Mlynská dolina CH1, Ilkovičova 6, 84215 Bratislava, Slovakia; 2Division of Environmental Physics, Faculty of Mathematics, Physics, and Informatics, Comenius University in Bratislava, Mlynská dolina F2, 84248 Bratislava, Slovakia

**Keywords:** cold plasma, yeast, *Saccharomyces cerevisiae*, mitochondria, oxidative stress, autophagy

## Abstract

Exposure of living cells to non-thermal plasma produced in various electrical discharges affects cell physiology and often results in cell death. Even though plasma-based techniques have started finding practical applications in biotechnology and medicine, the molecular mechanisms of interaction of cells with plasma remain poorly understood. In this study, the involvement of selected cellular components or pathways in plasma-induced cell killing was studied employing yeast deletion mutants. The changes in yeast sensitivity to plasma-activated water were observed in mutants with the defect in mitochondrial functions, including transport across the outer mitochondrial membrane (*∆por1*), cardiolipin biosynthesis (*∆crd1*, *∆pgs1*), respiration (*ρ*^0^) and assumed signaling to the nucleus (*∆mdl1*, *∆yme1*). Together these results indicate that mitochondria play an important role in plasma-activated water cell killing, both as the target of the damage and the participant in the damage signaling, which may lead to the induction of cell protection. On the other hand, our results show that neither mitochondria-ER contact sites, UPR, autophagy, nor proteasome play a major role in the protection of yeast cells from plasma-induced damage.

## 1. Introduction

Non-thermal plasma is an ionized gas typically produced by low-power electrical discharges in a gaseous environment. It contains multiple ions and radicals generated by reactions of electrons with gas molecules, which are highly reactive and can react with various biomolecules. The exposure of living cells to such plasmas, either directly or to the liquid media that have been treated with plasma, producing plasma-activated liquids, thus affects cell physiology and survival. By now, it has been well established that plasma and plasma-activated liquids can efficiently kill microorganisms, such as bacteria and fungi, and affect living cells, including animal or plant cells. Interactions of non-thermal plasmas with living cells have mostly been studied for the purpose of disinfection and sterilization or for the development of techniques by which specific cells, for example, cancer cells, can be selectively killed. The latter applications bring new potential in human medicine for the treatment of numerous pathologies, such as chronic wounds or cancer [1,2,3,4,5]. Although many studies have focused on interactions of cells with plasmas, the mechanisms involved in cell killing are still poorly understood, and the dissection of cellular pathways or structures, which play specific roles in plasma-induced cell killing, using a simple eukaryotic model is desirable.

Yeast *Saccharomyces cerevisiae* is a model organism with an unprecedented impact on the biology of eukaryotic cells. Easy cultivation and particularly its susceptibility to genetic analyses and manipulations, together with a broad range of methods for biochemical analyses, make it an ideal experimental system, in which the interactions of plasma with eukaryotic cells can be studied. When yeast cells are exposed to non-thermal plasma, they lose viability in a dose-dependent way (see [6] for review). Several mutations have been identified in yeast that increase the sensitivity of cells to the plasma treatment. These include deletions of genes encoding for either of the two superoxide dismutases [7], components of stress-responsive mitogen-activated protein kinase (MAPK) cascade [8], regulators of the cell cycle [8], and enzymes involved in the repair of double-strand breaks in DNA [9]. While superoxide dismutases apparently act upstream of the actual cell damage by inactivating reactive superoxide, the latter pathways participate in either signaling that results in the induction of cell protection or respond to the actual damage, e.g., by a repair of damaged DNA.

Although it has been reported that plasma treatment can trigger programmed cell death pathways such as apoptosis in mammalian cells [4,10,11,12,13], it does not appear that regulated cell death participates in the killing of yeast cells by plasma, as inactivation of genes encoding for components of these pathways in yeast does not affect the efficacy of plasma killing yeast cells [7]. Plasma-treated yeast cells die due to the damage resulting in a failure of some vital cellular activities rather than by the execution of the cell death program. This fact adds another advantage to using yeast as a model to study interactions of plasma with cells because the damage inflicted by plasma is not obscured by the changes resulting from the cell death program.

In this study, we tested the sensitivity of selected yeast deletion mutants to look for pathways related to the damage or cell protection from the damage caused by non-thermal plasma-activated water treatment.

## 2. Results and Discussion

To identify the cellular components or pathways that affect cell survival after treatment with non-thermal plasma, we tested selected *S. cerevisiae* mutants with individual deletions of genes encoding for components of selected pathways for their sensitivity to indirect treatment with plasma. In this experimental setting, the sterile deionized water was treated with plasma, producing the solution containing the reactive particles generated in the plasma, referred to as plasma-activated water (PAW). Cells were then incubated in PAW for a short period of time (60 and 120 min in this study), and the number of living cells was determined based on the ability of cells to form viable colonies on a complete growth medium (YPD).

When the wild-type strain was treated with PAW, roughly 50% of cells survived after 60 min (with respect to the cells surviving in the control untreated water taken as 100%), and an additional ~10% of cells died within the second hour of incubation in PAW, resulting in ~40% surviving cells after 120 min (Figure 1). The kinetics of the decrease of the number of viable cells in the treated suspension clearly correlates with the general idea of the decreasing concentration of short-lived reactive particles originating in PAW. At the same time, it reflects the presence of longer-lived reactive particles in PAW even after 60 min post plasma activation [14,15].

It should be noted here that even though the same conditions were used to generate PAW throughout this study, the individual batches of PAW differed slightly in their effectivity in cell-killing. The comparison of different strains was, therefore, possible among the strains treated with the same batch of PAW. The viability of individual mutants after treatment with PAW is therefore expressed relative to the viability of the wild type treated with the same batch of PAW in each particular experiment.

The set of deletion mutants selected for this study contained the mutants with deletions of nonessential genes involved in mitochondrial biogenesis and function, autophagy, and vacuolar metabolism and in response to various forms of stress.

### 2.1. Mitochondria

In yeast cells growing under various conditions, mitochondria are generally the main source of oxidative damage, and mitochondrial function is also required for resistance to oxidative stress [16]. The first set of mutants tested in this study, therefore, included mutants with defects in mitochondrial functions. These were mutants deficient in respiration and transport, such as the *ρ*^0^ mutant that is respiratory deficient due to the complete absence of mitochondrial DNA; mutants with a defect in ATP/ADP transport due to deletion of *AAC2* and *SAL1*, encoding for a major isoform of mitochondrial ADP/ATP carrier [17] and ATP-Mg/P_i_ transporter [18], respectively; a mutant lacking the outer membrane VDAC (*POR1*) [19,20]; as well as mutants with deletions of genes involved in mitochondrial biogenesis and maintenance, e.g., genes encoding for enzymes involved in the biosynthesis of cardiolipin (*CRD1*, *PGS1*) [21,22,23].

As shown in Figure 2, some of the tested mutants differed from the wild type in their sensitivity to PAW, with some strains being more sensitive and some strains being more resistant. Although no common pattern in these results can be observed, one can hypothesize on the role of each gene or related pathway in PAW-induced killing individually.

After the first hour of the PAW treatment, we observed increased survival of the *∆por1* mutant. The *POR1* gene encodes for the mitochondrial voltage-dependent anion channel (VDAC) [19,20]. This protein, also known as a mitochondrial porin, serves as the primary means of transport for low molecular compounds from the cytosol to mitochondria and vice versa. A potential explanation for the higher survival of cells lacking the VDAC could be a decrease in the ability of reactive particles originating from PAW to reach their targets in mitochondria. At the same time, the damage of mitochondrial targets would significantly participate in cell-killing by plasma. The latter is likely because it is in accordance with the earlier finding that mitochondrially localized superoxide dismutase (Sod2p) protects yeast cells from the plasma effects [7]. One can only assume whether or not the absence of VDAC in mitochondrial membranes may effectively limit the transport of plasma-generated reactive particles to mitochondria. The transport of other low molecular substrates to mitochondria is not totally absent in mitochondria devoid of VDAC as *∆por1* mutant is able to grow on non-fermentable carbon source at normal temperature, indicating that transport of ATP/ADP and substrates for mitochondrial respiration must occur at some level in this strain [24]. This residual permeability of the outer mitochondrial membrane, however, is profoundly decreased for some molecules, e.g., for NADH [25], indicating that the flow of metabolites through the outer mitochondrial membrane is limited.

Interestingly, the increased survival of the *∆por1* mutant is not observed after the second hour of treatment, indicating that the limited transport of reactive particles to mitochondria causes a lag in dying rather than net protection. Perhaps it may reflect the changes in the participation of different cellular targets at different times during the incubation in PAW as the composition of reactive particles in PAW changes in time: hydrogen peroxide and nitrites decrease, while nitrates increase after plasma activation [14,15].

The survival after treatment with PAW was also affected in two mutants lacking genes participating in the biosynthesis of cardiolipin. While both enzymes encoded by these genes, phosphatidylglycerol phosphate synthase (*PGS1*) and cardiolipin synthase (*CRD1*), are required for the final steps in cardiolipin biosynthesis and both deletion mutants, therefore, lack the cardiolipin in mitochondrial membranes [21,22,23,26], they differ significantly in the phenotype. The *∆pgs1* mutant is respiratory deficient. The *∆crd1*, on the other hand, has only mild growth defect on non-fermentable carbon source because it accumulates the cardiolipin biosynthesis intermediate–phosphatidyl glycerol—that can partially substitute for missing cardiolipin [27,28].

Here we observed a moderate increase in sensitivity to PAW in *∆crd1* but the opposite effect in *∆pgs1* (Figure 2). Although it may appear confusing that these two mutants differ in sensitivity to PAW, it correlates with the sensitivity to peroxides, which is increased in *∆crd1* cells while not in *∆pgs1* [29]. Because the defect in mitochondrial function is more severe in *∆pgs1* than in *∆crd1,* both these observations suggest that mitochondrial lipids by themselves, rather than mitochondrial functions supported by cardiolipin, are important in this situation. Increased resistance of *∆pgs1* can partially be explained by the fact that among mitochondrial lipids, cardiolipin is particularly susceptible to damage by reactive oxygen species [30,31]. This susceptibility mostly results from two factors – the high content of unsaturated fatty acyl chains, which are prone to oxidation, in cardiolipin, and from the localization of cardiolipin in the proximity of the respiratory chain, which is a prominent source of reactive oxygen species [30,31]. In the case of PAW-treated yeast cells, the latter does not apply because reactive particles, in this case, do not originate in mitochondria but enter the cell from the outside. The former would hold, but only partially, as yeast generally do not produce fatty acids with more than one carbon-carbon double bond [32,33]. The fatty acyl chains in cardiolipin in yeast are, therefore, mostly monounsaturated. What makes *∆crd1*, which lacks cardiolipin as well, more sensitive to PAW and to peroxides remains unclear. Perhaps the damage to phosphatidyl glycerol in the absence of cardiolipin is more toxic to cells than the oxidation of cardiolipin. Nevertheless, our results suggest that either cardiolipin or some mitochondrial functions that are dependent on the presence of cardiolipin play a role in damaging cells by plasma.

Increased cell survival was observed in *∆yme1* and *∆mdl1* strains (Figure 2). In these strains, the mitochondria lack the AAA-protease responsible for the degradation of unfolded or misfolded proteins [34,35,36] and the ABC transporter of the inner mitochondrial membrane that transports the resulting peptides from the mitochondrial matrix [37,38], respectively. It has previously been reported that *∆mdl1* mutant has increased resistance to oxidative stress, such as treatment with hydrogen peroxide [39]. Export of these peptides from mitochondria, thus, also likely affects pathways that protect cells from oxidative stress. The same pathways may be involved in the protection from damage by plasma treatment.

As further shown in Figure 2, increased sensitivity to PAW was observed in *ρ*^0^ strain—a mutant, which is devoid of mitochondrial DNA (Figure 2). Although the increased sensitivity may, in this case, result from the overall fragility and reduced robustness of the strain that lacks the functional oxidative phosphorylation, it may rather specifically reflect the requirement of functional respiratory competent mitochondria for resistance to oxidative stress [16]. The fact that a statistically significant decrease in the survival of this mutant was only observed after two hours of the treatment also indicates that functional mitochondria might be required for some of the stress response pathways that help cells survive in the later phases of the PAW treatment, perhaps because of changes in the chemical composition of PAW in time.

In contrast to *ρ*^0^, the survival of strains deficient in the transport of adenine nucleotides—*∆aac2* and *∆sal1*—did not significantly differ from the wild type. Since respiration is largely suppressed in the *∆aac2* strain due to the lack of ATP/ADP transport between mitochondria and cytosol, these results indicate that it is not the activity of the respiratory chain but rather its functionality that affects survival after treatment with PAW.

Other tested mitochondrial proteins included proteins involved in mitochondrial biogenesis and components of complexes at the contact sites of mitochondria and endoplasmic reticulum (ER), including *MMM1*, *MDM10*, *MDM12*, *MDM31*, *MDM32,* and *MDM34* (Figure 3). Deletion of neither of these genes affected the cell survival, with only a minute difference in the case of *∆mdm12*. This difference is, however, unlikely to result from the unfunctional complex of the mitochondria-ER contact site, as the absence of other subunits has no effect on cell survival.

### 2.2. Unfolded Protein Response

We further tested the participation of unfolded protein response (UPR) pathway. Although three UPR pathways have been described in mammalian cells, the pathway relying on Ire1p is the only known UPR pathway functional in yeast cells [40,41]. *IRE1* encodes for the protein kinase-endonuclease localized in the ER membrane that participates in the signaling of the accumulation of unfolded proteins in ER [42]. Another protein involved in the UPR, likely by affecting the Ire1 pathway, is ER membrane integral protein *Bxi1*. It is a homolog of mammalian Bax-inhibitor 1, which was first identified as a suppressor of the ability of mammalian proapoptotic protein Bax to induce cell death in yeast cells [43]. Later it was found that it is a calcium channel that mediates the release of calcium from ER to the cytosol [44,45]. Similar to the *∆ire1* strain, the *∆bxi1* mutant shows a decreased survival on media containing β-mercaptoethanol or tunicamycin, indicating that the function of Bxi1p supports UPR [46]. Signaling of ER stress by releasing Ca^2+^ into the cytosol is further mediated by a calcium and calmodulin-dependent protein phosphatase – calcineurin [47]. A regulatory subunit of calcineurin in yeast is encoded by *CNB1*.

As shown in Figure 4, neither of the tested mutants showed increased sensitivity to PAW with respect to the wild type. While the deletion of *BXI1* nor *CNB1* did not affect cell survival, *∆ire1* cells manifested an increased survival. This is rather surprising because it has been described previously that plasma treatment induces the Ire1 pathway in yeast [48], though it remained unclear whether this pathway effectively protects cells. As we show here, it appears that the activation of the Ire1 pathway not only does not help cells to survive after treatment with PAW, but the activity of this pathway, in fact, does make cells more vulnerable.

### 2.3. Autophagy and Proteasome

Autophagy is a cellular pathway that facilitates the removal of unwanted cell components by directing them to the vacuole for degradation. These cellular components include either the components that are no longer required during the process of adaptation to changes in the environment (e.g., removal of excessive mitochondria when cells are shifted from non-fermentable to fermentable carbon source or removal of peroxisomes when carbon source changes from fatty acids to sugar) or cellular components that have been damaged. The latter scenario contributes to survival under various stress conditions [49]. As during the exposure to PAW oxidative damage to multiple cell components likely happens, we also tested the survival of mutants with deletions of genes involved in autophagy or vacuolar metabolism (Figure 5). These included *ATG7*, encoding for a protein required for autophagosome formation; *VAC8*, required for the delivery to the vacuole; and *VAM7*, a vacuolar SNARE protein required for autophagy [50,51]. Additionally, we tested the participation of *TOR1*, a protein kinase, and *UTH1*, both of which are also involved in autophagy. Another means for removal of unwanted, e.g., damaged proteins, is the degradation in the proteasome. We, therefore, also tested mutants lacking components of the proteasome (*SEM1*, *RPN4,* and *RPN10*) [52,53,54,55,56,57].

Deletion of *VAC8* resulted in a strain slightly more sensitive to plasma treatment as compared to the wild-type strain; its survival rate is reduced by roughly 20% (Figure 5), both after 60 and 120 min of treatment. At the same time, the survival of the *∆atg7* and *∆vam7* strains did not significantly differ from the wild type. Since all three tested genes are required for autophagy, these results strongly suggest that autophagy does not play a significant role in maintaining survival after plasma treatment and that it must not be the defect in autophagy that affects the survival of the *∆vac8* mutant.

*TOR1* encodes for a protein kinase that is involved in the regulation of autophagy. *UTH1* was initially described as an aging gene, the deletion of which confers yeast cells a ‘longevity phenotype’ [58]. Interestingly, it has dual localization in the outer mitochondrial membrane and in the cell wall and participates in several unrelated processes, including stress response, mitochondria-specific autophagy, and cell wall function [59,60,61]. The survival of mutants with deletion of either of these genes did not differ from the wild type after 60 min of PAW treatment and was only slightly elevated (though with *p* = 0.0621 just below the threshold of statistical significance for *∆uth1*) after 120 min (Figure 5). If, in this case, the increase in survival reflects the defense of cells against PAW treatment, it may result from an inducible process that takes part in cell protection during the second hour of treatment. This may hypothetically involve autophagy or some other cellular process that would depend on the activity of Tor1p or Uth1p.

To test whether the autophagy is induced under conditions of plasma treatment, the cells of W303pho8∆60 strain were incubated in PAW for 1 h, washed with untreated water, and after 1 h, the autophagy was measured by alkaline phosphatase assay. In this assay, due to the *pho8∆60* mutation, the cells express an alkaline phosphatase precursor in the cytosol, which only gets activated when translocated to the vacuole by autophagy; hence the activity of alkaline phosphatase reflects the activity of autophagic machinery [62]. Autophagy in control cells was induced by starving for nitrogen. As shown in Figure 6, no increase in the alkaline phosphatase activity was detected in the plasma-treated cells, while we did observe it in control cells. In fact, the activity detected in PAW-treated cells was even lower than in the untreated cells. These results thus indicate that plasma treatment does not induce autophagy and that moderate differences in the sensitivity of *∆tor1*, *∆uth1* as well as *∆vac8* either result from their roles unrelated to autophagy or from some indirect effect.

As shown in Figure 5, some of the mutants with the deletions of genes encoding for the proteasome components (*SEM1*, *RPN4,* and *RPN10*) differed slightly in their sensitivity to PAW, with *∆sem1* being a little more resistant and *∆rpn4* a little more sensitive in the first hour of treatment while *∆rpn10* retained the sensitivity of a wild type. The lack of a clear trend toward the increased sensitivity in these mutants suggests that the proteasome does not play a significant role in protecting cells from the damage inflicted by plasma.

### 2.4. Growth Phase

Cells in different growth phases differ in several aspects, such as cell wall thickness, metabolism, or the activity of various cellular pathways affecting the resistance of cells to damage. As cells in different phases of growth may be differentially sensitive towards various agents, we further assessed the sensibility of cells growing exponentially and cells in the stationary phase of growth to PAW.

When exponentially growing yeast culture (strain CML282) is diluted in the pre-warmed YPD media, after 4 h, cells are exponentially growing, while after 24 and 48 h, they are in the earlier or later stationary phase, respectively. Cells grown under these conditions were treated with PAW for 1 h, and their viability was determined. As can be seen in Figure 7, we did not observe any differences between cells in different phases of growth. It thus appears that the growth phase does not influence the sensitivity of the yeast cells to PAW. This probably also indicates that the damage caused by plasma affects the processes equally essential in any phase of cell growth and that no growth-dependent cell protection pathway significantly influences the ability of cells to repair the damage caused by PAW.

Taken together, among the mutants whose sensitivity to the treatment with PAW was tested in this study, we identified several that manifested the changes with respect to the wild type. A considerable subset of these mutants were the ones with the defect in mitochondrial functions, including transport across the outer mitochondrial membrane (*∆por1*), cardiolipin biosynthesis (*∆crd1*, *∆pgs1*), respiration (*ρ*^0^), and assumed signaling to the nucleus (*∆mdl1*, *∆yme1*). Although one has to bear in mind that reactive particles originating from PAW do likely target countless cellular targets nonspecifically, collectively, these results indicate that mitochondria play a substantial role in PAW effects as both a target of the damage, and the participant in the damage signaling, which may lead to the induction of the cell defense.

Our results also show that neither mitochondria-ER contact sites, UPR, autophagy, nor proteasome play major roles in the protection of cells from plasma-induced damage in yeast cells.

Since the yeast cell components, whose roles in plasma-induced damage were investigated in this study, have homologs in mammalian and other eukaryotic cells, it is reasonable to assume that similar also applies to mammalian cells, in which some of these effects may be hard to detect due to the activation of programmed cell death pathways.

## 3. Materials and Methods

### 3.1. Strains and Growth Conditions

The yeast strains used throughout this study were CML282 (*MAT**a** ura3-1*, *ade2-1*, *leu2-3,112*, *his3-11,15*, *trp1-Δ2*, *can1-100*, *CMV_p_(tetR-SSN6)::LEU2*) kindly provided by Enrique Herrero, Universitat de Lleida [63] and BY4742 (*MAT**α** his3Δ1*, *leu2Δ0*, *lys2Δ0*, *ura3Δ0*). The mutant strains with deletions of specific genes were either obtained from EUROSCARF collection or prepared by standard gene replacement techniques. Briefly, disruption cassettes containing selection marker gene kanMX4 flanked by sequences corresponding to the 5’ and 3’ untranslated regions of the target genes were prepared by polymerase chain reaction (PCR) using plasmid pFA6a-kanMX4 [64] as a template. Yeast was transformed by standard lithium acetate protocols. After the transformation of the wild-type yeast strain, transformants were selected by growth on selective plates containing 200 μg/mL geneticin (G418). Disruptions of specific genes were verified by PCR, using oligonucleotide primers annealing upstream and downstream of respective genes and inside the marker gene.

The strain W303pho8∆60 (*MAT**a** ura3-1*, *ade2-1*, *leu2-3,112*, *his3-11,15*, *trp1-Δ2*, *can1-100*, *pho8::PHO8∆60-URA3*) [60] was used for measuring the autophagy.

Cells were cultivated on complete YPD (1% yeast extract, 2% peptone, and 2% glucose) media at 28 °C.

### 3.2. Plasma-Activated Water Generation

The experimental setup used for the plasma treatment of water (PAW generation) is depicted in Figure 8. A high DC voltage was applied on a hollow needle electrode through a ballast resistor (10 MΩ). The transient spark discharge was operated in a point-to-plain geometry between the tip of the needle electrode and a grid electrode 10 mm apart. The discharge parameters were measured by a 200 MHz digitizing oscilloscope (TDS 2024, Tektronix Inc., Beaverton, OR, USA) using a high voltage probe (Tektronix P6015A, Beaverton, OR, USA) to record voltage, while the current was measured on a 1 Ω resistor. The frequency of repetitive spark discharge pulses was maintained at 1 kHz. More details on the transient spark discharge regime and typical parameters can be found in [65]. The treated deionized water was pumped through the needle electrode and electrosprayed through the plasma discharge by a syringe pump with a constant flow rate of 1 mL/min.

Freshly prepared PAW was used for the treatment of cells. Alternatively, PAW was frozen immediately after preparation, stored at −80 °C, and used instantly after thawing. This freezing was applied to preserve the concentrations of reactive particles and the antimicrobial activity of PAW, which otherwise decay within a few hours post plasma treatment [14,15,66].

### 3.3. Cells Treatment and Viability Assay

Cells were grown to the exponential phase (unless stated otherwise) and washed with distilled water. Cell suspensions of 5 × 10^7^ cells/mL in sterile deionized water were diluted tenfold in PAW or untreated deionized water (control) and incubated for indicated time periods. In each experiment, the same batch of PAW was used for the treatment of the mutant and control (wild-type) strains. Samples of the cell suspension after the indicated times of plasma treatment (60 min and 120 min) and control untreated samples were diluted in fresh (non-treated) water and spread on the Petri dishes containing complete growth media (YPD; 1% yeast extract, 2% peptone, 2% glucose, 2% agar).

The viability of treated cells was evaluated after 2–3 days of cultivation at 28 °C as a ratio of the number of colonies observed at the plate with PAW-treated cells relative to the number of colonies formed from untreated control cells. To compare the viability of different strains, the relative viability was calculated as a ratio of the viability of an individual strain to the viability of a wild-type strain. Shown are the results of at least three independent experiments.

Data were analyzed using the GraphPad Prism 9 software (GraphPad Software Inc., San Diego, CA, USA). The statistical significance of the decrease in viability of wild-type strain was assessed by one-way analysis of variance (ANOVA) with Tukey’s multiple comparison test. The statistical significance of differences between the individual mutants and the wild type was assessed by a two-way ANOVA with Šidaks’s multiple comparison test. Adjusted *p*-values obtained from multiple comparison tests of *p* ≤ 0.05, *p* ≤ 0.01 and *p* ≤ 0.0001 are indicated in the graphs. The *p*-value of 0.05 was considered a threshold for statistical significance.

### 3.4. Monitoring of Autophagy

Autophagy was monitored by an alkaline phosphatase (ALP) assay [62]. A yeast strain W303pho8∆60 was cultivated in YPD and incubated for 1 h either in PAW or in untreated water (negative control), centrifuged, and further incubated in untreated water for 1 h. As a positive control, autophagy was induced in the same strain by cultivation under conditions of nitrogen starvation—16 h in YNB without ammonium sulfate and without amino acids (2% glucose, 0.17% yeast nitrogen base). Alkaline phosphatase activity was then measured spectrophotometrically as absorbance at 420 nm, using p-nitrophenyl phosphate (Sigma, St. Louis, MO, USA) as a substrate.

## 4. Conclusions

We have examined the effect of non-thermal plasma-activated water on the survival of yeast *S. cerevisiae*. We identified several genes, the deletion of which results in altered sensitivity to PAW. Although the effect of PAW (and likely of non-thermal plasma in general) is pleiotropic and includes the chemical damage of multiple cellular targets, our results suggest that some of the cellular targets may be particularly vulnerable. These likely involve mitochondria since, in several mutants in mitochondrial components, the survival of the treated cells is affected. In the tested mutants with deletions of genes encoding for other than mitochondrial components that affected the sensitivity to PAW, the change in survival can generally be attributed to the altered cell signaling that affects the cells’ response to stress.

On the other hand, we have shown that neither UPR nor autophagy play significant roles in PAW resistance and that the cell sensitivity to plasma does not change with the growth phase of the yeast culture.

## Figures and Tables

**Figure 1 ijms-24-09391-f001:**
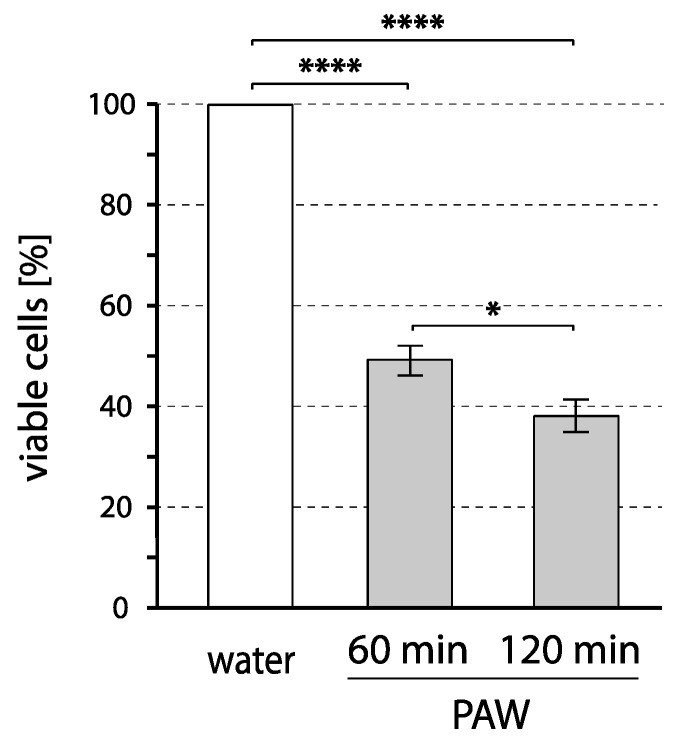
Effect of PAW on the viability of wild-type yeast strain. Cells of the wild-type strain (BY4742) were incubated with PAW or untreated water (control). After the indicated time of incubation (60 or 120 min) at laboratory temperature, aliquots were spread onto Petri dishes with complete growth medium (YPD). The proportion of living cells was evaluated after 2–3 days of cultivation at 28 °C. Plotted values represent the number of colonies formed by treated cells relative to the untreated control, with the control normalized to 100%. Mean values and standard errors of proportion are shown for data from five independent experiments. Asterisks indicate a statistical significance of *p* < 0.01 (*) and *p* < 0.0001 (****).

**Figure 2 ijms-24-09391-f002:**
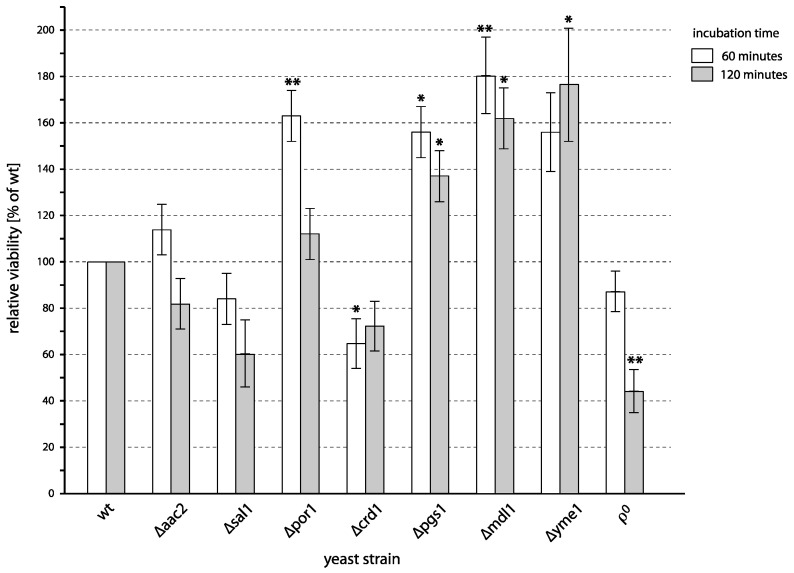
Viability of yeast mutants defective in selected mitochondrial functions after incubation with PAW. Cells of different deletion mutants were incubated with PAW or untreated water for 1 and 2 h, and viability was determined by plating on YPD. The viability of each mutant (the number of colonies formed by the PAW-treated cells to the number of colonies formed by the untreated control) is expressed relative to the viability of the wild type (shown here as 100%). Mean values and standard errors of proportion are shown for data from 3 independent experiments. Asterisks indicate a statistical significance of *p* < 0.05 (*) and *p* < 0.01 (**) of differences between mutants and the wild type.

**Figure 3 ijms-24-09391-f003:**
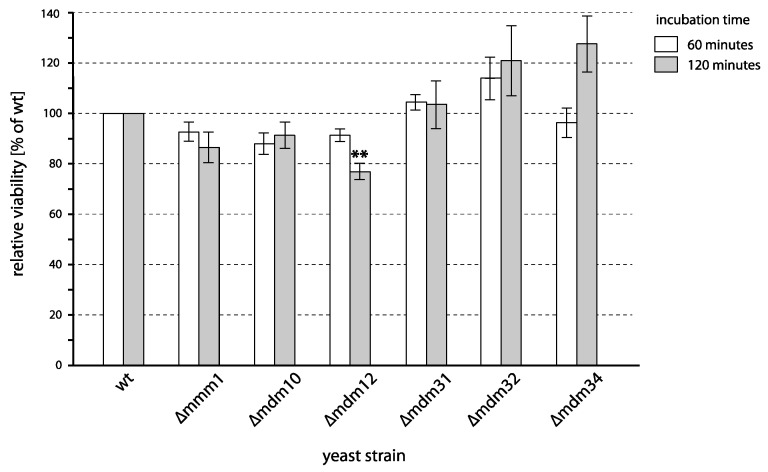
Viability of yeast mutants defective in mitochondrial inheritance and mitochondria-ER contact sites after incubation with PAW. Cells of different deletion mutants were incubated with PAW or untreated water for 1 and 2 h, and viability was determined by plating on YPD. The viability of each strain is expressed relative to the viability of a corresponding wild type (shown here as 100%). Mean values and standard errors of proportion are shown for data from three independent experiments. Asterisks (**) indicate a statistical significance (*p* < 0.01) of the differences between mutants and the wild type.

**Figure 4 ijms-24-09391-f004:**
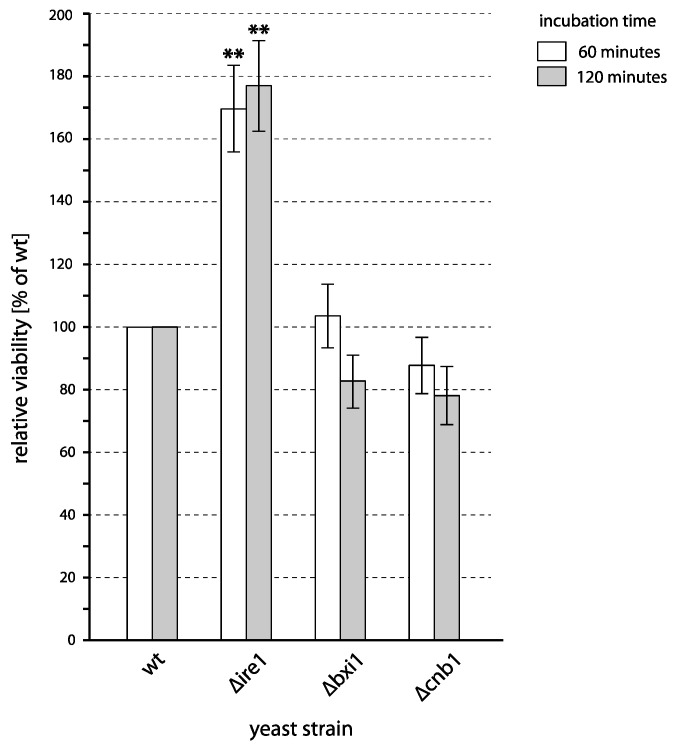
Viability of yeast mutants defective in UPR after incubation with PAW. Cells of different deletion mutants were incubated with PAW or untreated water for 1 and 2 h, and viability was determined by plating on YPD. The viability of each strain is expressed relative to the viability of a corresponding wild type (shown here as 100%). Mean values and standard errors of proportion are shown for data from three independent experiments. Asterisks (**) indicate the statistical significance (*p* < 0.01) of the differences between mutants and the wild type.

**Figure 5 ijms-24-09391-f005:**
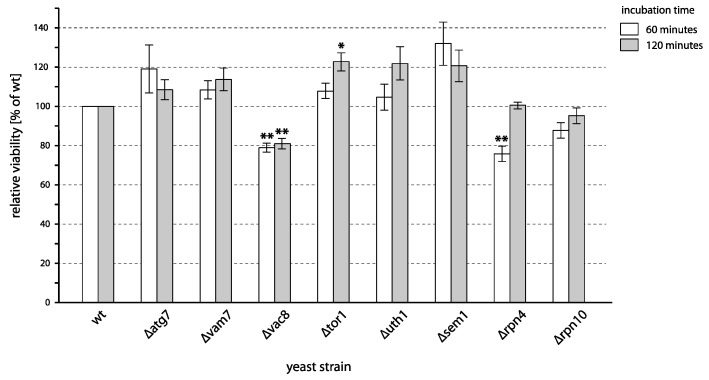
Viability of yeast mutants defective in autophagy and proteasomal protein degradation after incubation with PAW. Cells of different deletion mutants were incubated with PAW or untreated water for 1 and 2 h, and viability was determined by plating on YPD. The viability of each strain is expressed relative to the viability of a corresponding wild type (shown here as 100%). Mean values and standard errors of proportion are shown for data from 3 independent experiments. Asterisks indicate a statistical significance of *p* < 0.05 (*) and *p* < 0.01 (**) of differences between mutants and the wild type.

**Figure 6 ijms-24-09391-f006:**
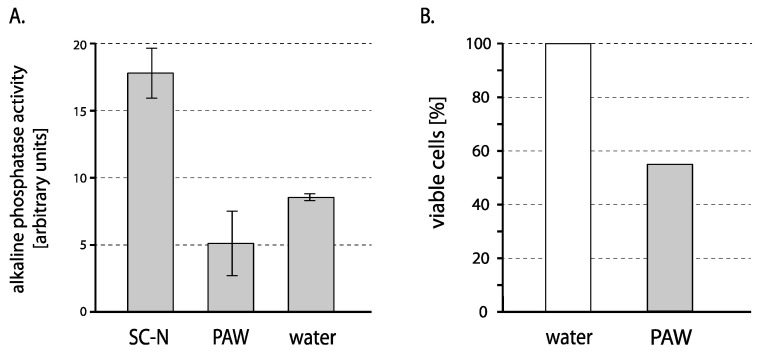
Autophagy activity in PAW-treated cells. (**A**) In cells incubated in PAW (PAW), cells incubated in water (dH_2_O, negative control), and cells cultured under nitrogen starvation conditions (SC-N, positive control), autophagy was quantified by measuring the activity of a reporter — alkaline phosphatase. Shown is a representative of three experiments. (**B**) The viability of PAW-treated cells in this experiment is expressed as a percentage of viable cells detected by plating on YPD (PAW-treated vs. untreated cells in water taken as 100%).

**Figure 7 ijms-24-09391-f007:**
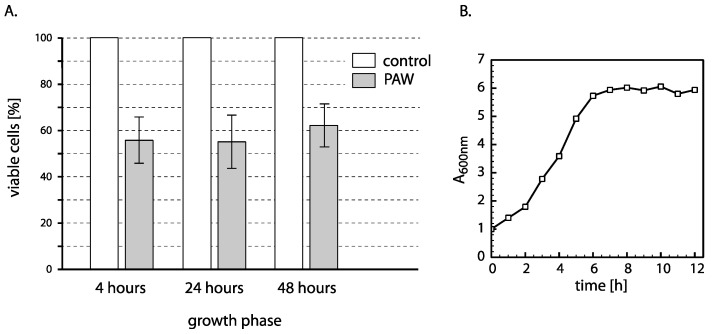
Viability of yeast cells in different growth phases after incubation with PAW. (**A**) Cells at different growth stages were incubated with PAW (grey bars) or untreated water (white bars) for 1 h, and the viability was determined by plating on YPD. Viability is expressed as the percentage of the number of colonies formed by PAW-treated cells from the number of colonies formed by untreated control (taken as 100%). (**B**) The growth curve of CML282 at the conditions of the experiment.

**Figure 8 ijms-24-09391-f008:**
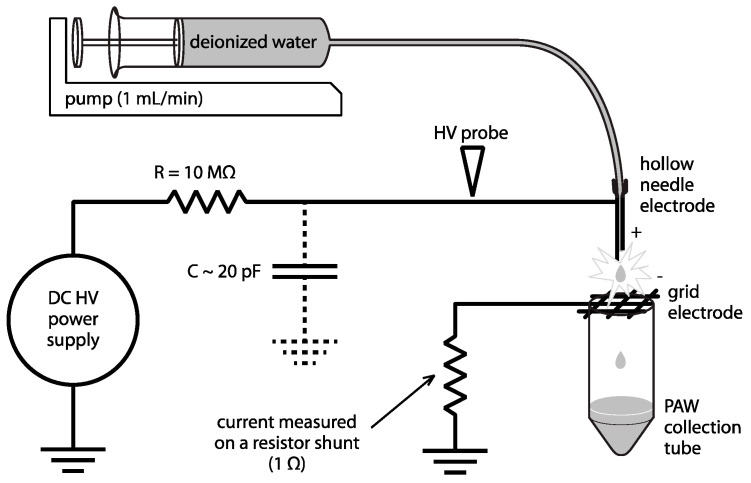
Experimental setup for plasma treatment of water. A DC high voltage was applied on a hollow needle electrode through a ballast resistor (10 MΩ) and a high voltage cable representing the capacity of ~20 pF. The transient spark discharge was operated between the tip of the needle electrode and a grid electrode (10 mm apart). The parameters of the discharge were measured by a digitizing oscilloscope, and frequency was maintained at 1 kHz. Treated deionized water was pumped through the needle electrode and electrosprayed through the plasma discharge by a syringe pump with a constant flow rate of 1 mL/min.

## Data Availability

Not applicable.

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
