# Peer review of "Defects in Mitochondrial Functions Affect the Survival of Yeast Cells Treated with Non-Thermal Plasma"

_ijms, 2023, doi:10.3390/ijms24119391_

Round 1
Reviewer 1 Report
In general, the reviewer of the manuscript had an opposing impression. On the one hand, there is rich data here; on the other hand, the presentation and interpretation of the results obtained by the authors is not always successful.
This manuscript is hard to read because of the poor quality of the language; this is noticeable to a reviewer who is not a native English speaker. Manuscript needs substantial English language editing; the authors use expressions in the text, the context of which is often not suitable for description in this manuscript. For example, in the "Abstract" section (lines 20-22), the sentence starts with the word "Collectively" when the context would probably read "In general/total". And in the same place, lines 20-21, the authors write "... mitochondria play a substantial role in plasma-activated water cell killing, as both a target of the damage ...". Maybe it should be "...mitochondria play an important role in killing cells with plasma-activated water, both as a target of damage..."?
The purpose of this study is not clear from the "introduction" section. The section itself is written somewhat chaotically and consists of several fragments of text that are not connected by a single meaning; it seems that one could speculate about the direct or indirect influence of nonthermal plasma on cell components (for example, the same mitochondria), or discuss in more detail the role of defects in cellular components (mitochondria again) in changing the sensitivity of cells to active radicals, etc. But this is not the case, and why it was necessary to carry out this study is not clear.
Section "materials and methods".
The content of this section needs to be improved as it does not allow for a clear presentation of the design of the experiments.
Why did the authors freeze plasma-treated water? If this was later used in the experiment, then why is nothing said about it in the “Results” section? How many times were the experiments duplicated? What statistical methods did the authors use?
As it stands, subsection 3.1 does not provide insight into the strain transformation methods used and associated procedures; authors should describe this subsection in more detail.
Section "Results and Discussion".
The reviewer believes that it is not entirely correct to measure the survival of mutant strains as a percentage of the wild type of the strain (as in Figures 1-4), since it is generally accepted that survival is a value that shows how much the number of cells has changed after any treatment compared to original. In the experiments, the authors used untreated water as a control, obviously it would be more correct to make comparisons with this control. It seems that the presentation of the figures should be changed in such a way that readers understand how the survival of mutant and wild-type strains changes. It may be that due to a different representation of the data, the results will look different; this will entail the need for changes in the discussion of the results and, accordingly, the conclusions.
One more question: active radicals, which are formed during plasma treatment, have a short life span. Why did the authors incubate yeast cells with treated water for 120 minutes?
The text fragment in lines 74-79 should be deleted or moved, as it is a description of the research methodology and duplicates similar text from the "materials and methods" section.
Reviewer 2 Report
In Strizova et al., the authors studied the effects of non-thermal plasma on survival of yeast cell by employing various yeast mutants involved in mitochondrial functions, cardiolipin biosynthesis, respiration, autophagy etc.
I do not think the present manuscript does not have the quality data and lacks scientific merit. The manuscript appears more like a review article than a research article. The only experiment the authors conducted in this manuscript is the survivability of different yeast strains upon treatment with non-thermal plasma and an additional experiment of alkaline phosphatase assay for which the conclusions do not reflect the results. Although the experiment was done at two time points, they ignore to explain the contrasting results observed at two different time points (por1 mutant). No statistical testing. As the differences between the control and tested groups are not of higher magnitude, the statistical testing is compulsory to infer whether the differences are of significant. Further, the writing is very hard to comprehend the results and at times the data and results are in contrast.
Other comments:
The authors discuss more the literature of the different genes rather than on their results all through the paper.
Need to show the data for WT – with treatment and without treatment.
Each of mutants should be complemented with wild type copy of gene.
Some of the mutants are not even discussed in the results eg aac2, sal2.
The quality of english is difficult to comprehend the results. It should be edited thoroughly.
Reviewer 3 Report
In this study, the researchers investigated the molecular mechanisms involved in the killing of living cells when exposed to non-thermal plasma, which is commonly used in biotechnology and medicine. To do this, they used yeast deletion mutants and observed changes in yeast sensitivity to plasma-activated water.
The researchers found that mutants with defects in mitochondrial functions, such as transport across the outer mitochondrial membrane (Δpor1), cardiolipin biosynthesis (Δcrd1, Δpgs1), respiration (ρ0), and signaling to the nucleus (Δmdl1, Δyme1), were more sensitive to plasma-activated water. This suggests that mitochondria play a significant role in both the damage caused by the plasma and the signaling of that damage, which may lead to the induction of cell protection.
However, the researchers found that neither mitochondria-endoplasmic reticulum contact sites, the unfolded protein response, autophagy, nor the proteasome played a major role in protecting yeast cells from plasma-induced damage. Overall, this study highlights the importance of mitochondrial function in the response of cells to non-thermal plasma and provides insights into the molecular mechanisms underlying this process.
A specific comment is below.
1. In this paper, authors showed the survival of each mutant strain after exposing PAW and significant effects for some mutant strains. The reviewer concerns that the gene deletion already affects the survival rate of mutant yeasts under normal conditions. Authors should compare the survival rates of each deletion mutant with WT yeast under normal conditions. This would help clarify whether the observed effects of PAW exposure on mutant strains are due to gene deletion.
Round 2
Reviewer 1 Report
The authors have done significant work, which resulted in improving the quality of this manuscript.
However, there are gross errors in the text of the manuscript, which, from the point of view of the reviewer, require mandatory correction; this concerns the negligent attitude of the authors to the data obtained and their processing, including statistical procedures, visualizations, etc.
Section "results".
Figure 1 captions do not correspond to the type of data presented here. The vertical axis indicates survival as a percentage, that is, it is a question of the proportion of survivors; the caption says "Plotted values represent the number of colonies formed by treated cells relative to the untreated control". In the same place, it is indicated about the mean values and standard deviation, although it is generally accepted for proportions of features to use the error of the proportion. There are also questions about the validity of the differences between the data. From Figure 1, it becomes apparent that there are (possibly) significant differences between the control and experimental data. However, it is indicated that there is a significant difference between the experimental data (60 and 120 minutes) at a high level, although the difference between them is about 10 percentage points. Is it really? What criteria were used for comparison?
In figures 2, 3, 4, 5 the same problem. In addition, it is not clear here between which data there are significant differences; it seems that it is necessary to indicate in the captions to the figures between which data groups there was a comparison.
In the text of the manuscript, there are only results about freshly prepared water that was treated with plasma, although in the section "Materials and Methods" there is a mention of the use of frozen water in the experiment.
Section materials and methods. The reviewer believes that the authors should, firstly, describe in a separate subsection all those statistical methods and criteria that they used; secondly, to clarify these statistical tests, since it is not entirely clear to the reviewer why the authors used a paired t-test to compare proportions (lines 455-456).
Reviewer 3 Report
The authors have addressed my queries satisfactorily.
Author Response
Authors thank the reviewer.
Round 3
Reviewer 1 Report
The reviewer thanks the authors for their attentive attitude to questions and comments regarding the submitted manuscript.
Although this manuscript may be accepted in its present form as is, the reviewer believes that it is not correct to refer to unpublished results. The authors have no evidence (in the text of the manuscript) that plasma-treated frozen and freshly prepared water samples do not differ from each other in properties. Perhaps it would be right to either add the results of a study where the properties of frozen or freshly prepared water samples would be compared, or completely remove about frozen water. Or it might be better to refer to some study showing that quick freezing at -80 ensures the preservation of active radicals that were formed during the treatment of water with plasma.
In general, the reviewer does not believe that this remark should prevent the acceptance of the manuscript for publication. The reviewer leaves the decision on the similarity or differences between frozen and freshly prepared water samples to the discretion of the authors.
Good luck.
Author Response
Dear Reviewer,
thank you very much for your review.
As our result confirming the activity of forzen/thawed PAW will be a part of another study, we would prefere not to include them in this paper. We have, however, found a published study, in which the antimicrobial activity of frozen PAW was investigated, that confirms are unpublished results and shows that the activity of PAW can be preserved by freezing at -80°C. We have included it as the additional reference in the manuscript (reference no. 66).
Yours sincerely,
Peter Polčic